# Modeling habitat distribution and niche overlap of Asian horseshoe crabs: Implications for conservation

Jian Liao[1], Chun-Hui Xiong[2], Gao-Cong Li[3], Jia-Yu Li[1], Yuan-Feng Yang[1], Shui-Yuan Zhang[1], Yi-Yang Li[1], Kai-Lin Zeng[1], Mei-Ling Hu[1], Yu-Song Guo[1], Zhong-Duo Wang [1,4]*

**1** Key Laboratory of Aquaculture in South China Sea for Aquatic Economic Animals of Guangdong Higher Education Institutes, College of Fisheries, Guangdong Ocean University, Zhanjiang, China, **2** Guangdong Provincial Field Observation and Research Station for Marine Ecosystem in Hanjiang River Estuary - Nanao Island Area, Guangzhou, China, **3** College of Electronics and Information Engineering, Guangdong Ocean University, Zhanjiang, China, **4** Guangdong Provincial Key Laboratory of Aquatic Animal Disease Control and Healthy Culture, College of Fisheries, Guangdong Ocean University, Zhanjiang, China

* wangzd@gdou.edu.cn

## Abstract

Asian horseshoe crabs are ancient organisms essential for the balance of marine ecosystems. However, detailed information on their ecology and the environmental factors influencing their distribution remains limited. In this study, we analyzed habitat characteristics, potential distribution, and niche overlap for three species: *Tachypleus tridentatus*, *Carcinoscorpius rotundicauda*, and *Tachypleus gigas*. Predictive modeling using MaxEnt and niche analysis revealed that water depth and distance to land are key factors determining species distribution, with species-specific environmental influences: *T. tridentatus* is affected by maximum summer chlorophyll-a, *C. rotundicauda* by minimum chlorophyll-a, and *T. gigas* by wind speed. In terms of niche overlap, the highest degree of overlap was observed between *C. rotundicauda* and *T. gigas*, while the overlap between *T. tridentatus* and *T. gigas* was the lowest. The results highlight priority conservation areas, providing insights for management and protection strategies amid current environmental threats.

## Introduction

Horseshoe crabs, as ancient creatures, have fossils dating back to the Ordovician period of the Palaeozoic Era approximately 475 million years ago [1–3]. Their morphology has remained virtually unchanged for over 200 million years, playing a fundamental role in marine ecosystems and the maintenance of biodiversity [4,5]. Globally, there are four species of horseshoe crabs: Chinese horseshoe crab *Tachypleus tridentatus*, mangrove horseshoe crab *Carcinoscorpius rotundicauda*, coastal horseshoe crab *Tachypleus gigas*, and American horseshoe crab *Limulus*

**Data availability statement:** Data used in the study have been included in the article/ Supporting information. For detailed information, please refer to the attached tables: Tachypleus tridentatus.XLS, Tachypleus gigas. XLS, and Carcinoscorpius rotundicauda.XLS.

**Funding:** This study was funded by the Guangdong Provincial Field Observation and Research Station for Marine Ecosystem in Hanjiang River Estuary - Nanao Island Area Open Fund (Grant No. HNS202407), the National Key Research and Development Program of China (Grant No. 2024YFD2401803 & 2024YFD2401802), the Guangdong Provincial Ordinary University Youth Innovative Talent Project in 2024 (Grant No. 2024KQNCX134), the Guangdong Provincial Special Fund Project for Talent Development Strategy in 2024 (Grant No. 2024R3005), and the Guangdong Ocean University Scientific Research Startup Funding Project (Grant No. 060302022312).

**Competing interests:** The authors have declared that no competing interests exist.

*polyphemus* [4,6], and all of them have been listed in the IUCN Red List of Threatened Species [7,8]. Among them, *T. tridentatus*, *C. rotundicauda*, and *T. gigas* are primarily distributed in Asian seas and are therefore also referred to as Asian horseshoe crabs [4]. In contrast, *L. polyphemus* is distributed in the Americas and is thus called the American horseshoe crab [9]. *T. tridentatus* exhibits a broad geographical distribution across East Asia and represents the species with the largest latitudinal range among all horseshoe crabs [4]. *C. rotundicauda* primarily inhabits the unique mangrove ecosystems of tropical seas and is highly dependent on these ecosystems [10]. *T. gigas* is widely distributed in tropical and subtropical seas of Southeast Asia, particularly in Vietnam, the Philippines, Indonesia, and other regions, often residing in intertidal sandy or muddy beaches [11].

However, with the continuous expansion of human activities and the sustained deterioration of marine environmental quality, these ancient and precious Asian horseshoe crab populations are facing unprecedented survival crises [5,12,13]. From shallow-sea fishing to deep-sea resource exploitation, human exploitation of marine resources has directly led to sharp declines in numerous marine species populations, including Asian horseshoe crabs [12,14]. Overfishing not only reduces their direct populations but also disrupts the integrity of their food chains, affecting the natural recovery ability of the populations [15]. Meanwhile, the habitats of Asian horseshoe crabs have also suffered severe damage [8]. With the acceleration of urbanization, large amounts of beaches and wetlands, particularly nearshore spawning grounds, have been developed for construction, transforming these ideal habitats for Asian horseshoe crabs into inaccessible forbidden zones [8]. Additionally, waste generated from industrialization and agricultural activities in coastal areas is directly discharged into the ocean without effective treatment, not only polluting water quality but also damaging the underwater ecological environment, further compressing the living space of Asian horseshoe crabs. Therefore, conducting in-depth research on the potential habitats of the three Asian horseshoe crab species (*T. tridentatus*, *C. rotundicauda*, and *T. gigas*) and the important ecological factors affecting their distribution is crucial for formulating scientific and reasonable conservation strategies, promoting population recovery, and protecting biodiversity.

Species distribution models are an effective method to help researchers gain a deeper understanding of the intricate relationship between species and their environments [16,17]. Among them, the MaxEnt model based on the principle of maximum entropy integrates environmental variables and species distribution data to construct a probability model of species distribution [18,19]. The core advantage of this model lies in its strong predictive ability and efficient use of data [20]. Compared to other models, MaxEnt has relatively relaxed requirements for data, and can make accurate predictions with only a small amount of species distribution data and environmental factor data [21]. In addition, the MaxEnt model can also handle environmental factors with nonlinear relationships and fully consider the spatial heterogeneity of species distribution, thereby effectively predicting the potential distribution areas of species under different environmental conditions [19,22]. After more than a decade of development, the MaxEnt model, as the mainstream method in species distribution

modeling, has been systematically and maturely applied in the field of terrestrial biology, and is entering a rapid development stage in the field of marine biology [23–25]. For example, the MaxEnt model has been used to simulate scad fish (*Decanters* spp.) in the northern South China Sea [26], and joint species distribution models have been employed to predict the distribution of seafloor taxa and identify vulnerable marine ecosystems in New Zealand waters [27]. Regarding research on horseshoe crab habitat suitability and suitability relationships, Chinese scholars have conducted local studies in the Beibu Gulf region [28]. However, the environmental factors studied were limited to climatic factors, which have a weak correlation with horseshoe crabs, a marine benthic organism [28]. Our study, starting from the marine environment, comprehensively considers physical, chemical, and nutritional factors, which are more in line with the habitat conditions of horseshoe crabs, a marine benthic organism, and offers predictable advantages.

As horseshoe crabs are typical intertidal benthic organisms, we hypothesize that water depth, land distance, and specific environmental factors are the primary determinants of the distribution of the three Asian horseshoe crab species. We use the MaxEnt model to predict potential habitats and analyze niche overlap, aiming to provide insights for horseshoe crab conservation. Meanwhile, this study will also provide methods and ideas for the conservation research of other endangered species, promoting the development of global biodiversity conservation efforts.

## Materials and methods

### Species occurrence sites

The distribution point data from adults for *T. tridentatus*, *C. rotundicauda*, and *T. gigas* primarily originated from the Global Biodiversity Information Facility (GBIF, website: https://www.gbif.org/), the Ocean Biodiversity Information System (website: https://obis.org/), and literature reviews. During the data preprocessing phase, we carefully cross-checked each occurrence record against the known distribution ranges of *T. tridentatus*, *C. rotundicauda*, and *T. gigas*, in addition to removing duplicates and anomalous points, records with inaccurate coordinates or located in terrestrial environments were manually verified and corrected whenever possible [21]. To further reduce spatial autocorrelation and data redundancy, we adopted a grid system method, dividing the study area into 5km×5km grid cells. Within each grid cell, we retained only the occurrence point closest to the geometric center of the grid, thereby reducing the density of the dataset while preserving its key spatial distribution characteristics, effectively avoiding issues of overfitting or bias [29]. After rigorous screening and spatial analysis, we finally compiled datasets containing 268 coordinate records for *T. tridentatus*, 305 for *C. rotundicauda*, and 231 for *T. gigas* (Fig 1).

### Environmental variables

To analyze the distribution patterns of three Asian horseshoe crab species, *T. tridentatus*, *C. rotundicauda*, and *T. gigas*, we downloaded 48 marine environmental variables from the Global Marine Environment Dataset (GMED, https://gmed.auckland.ac.nz/index.html). These variables were categorized into 26 physical variables, 9 chemical variables, and 13 nutritional variables, as detailed in Table 1, with a resolution of 2.5 minutes of latitude/longitude. Subsequently, we utilized the "Extract Multi Values to Points" tool in the ArcGIS toolbox to extract the values of these environmental variables from the distribution points of the three Asian horseshoe crab species (S1–S3 File). Following that, in the R programming language, we employed the *usdm* package (version 1.1–18) to conduct Variance Inflation Factor (VIF) analysis, setting the VIF threshold at <10 to eliminate factors with multicollinearity, as recommended in the study by Naimi et al. (2014).

### Potential habitat and niche analysis

In this study, we employed the Maximum Entropy (MaxEnt) modeling approach, integrating distribution point data with a multitude of marine environmental variables (including physical, chemical, and nutritional factors), to predict the suitable habitat distribution of *T. tridentatus*, *C. rotundicauda*, and *T. gigas* using MaxEnt version 3.4.1 [19]. The 75/25 (training/

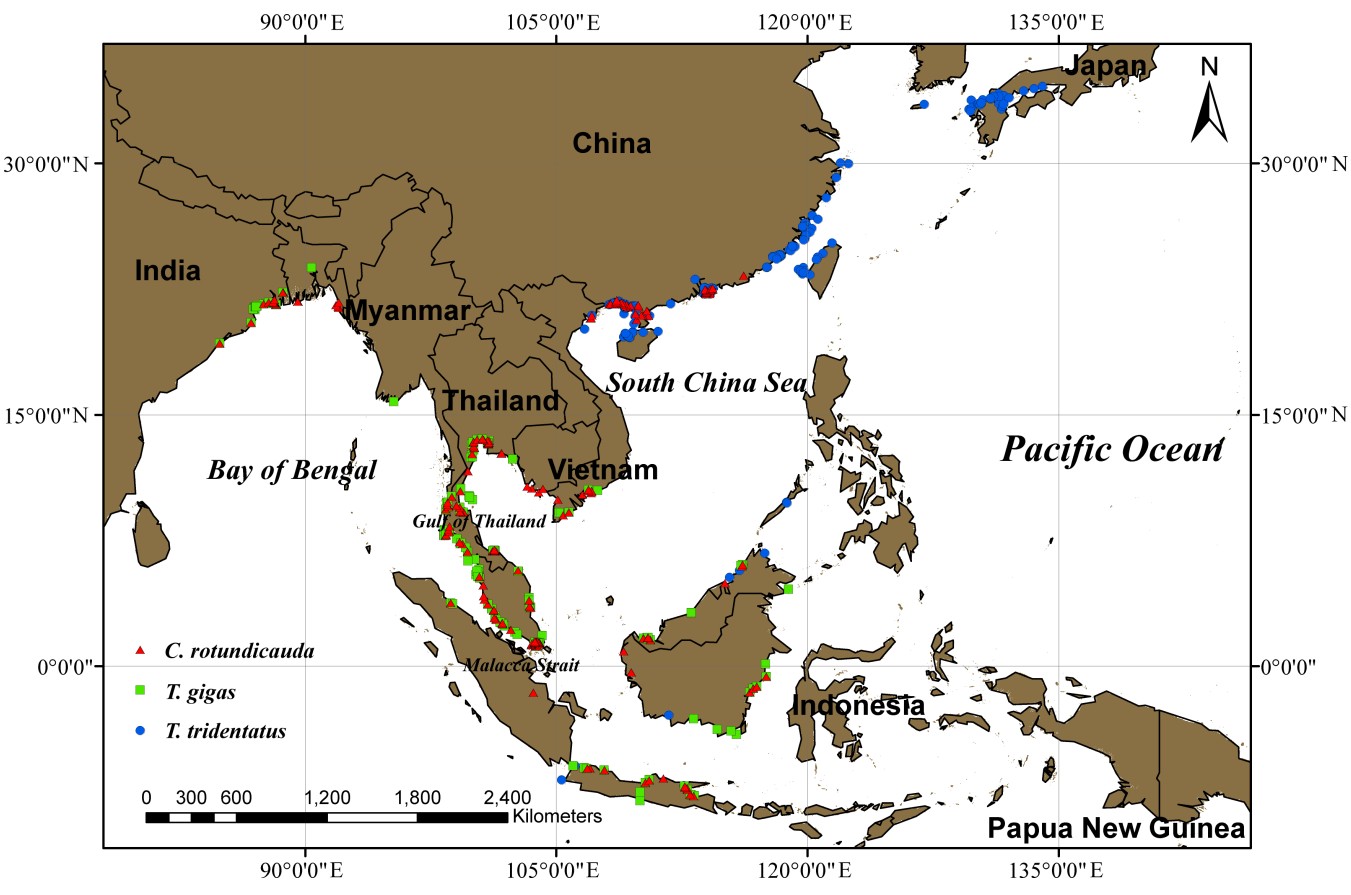

**Fig 1. Distribution Maps of *T. tridentatus*, *C. rotundicauda*, and *T. gigas*.**

testing) split was chosen as it is widely used in ecological modeling studies, ensuring a balance between adequate model training and robust validation. This process was repeated ten times to enhance reliability [20]. MaxEnt parameters were adjusted to minimize overfitting, using a regularization value of 1, a maximum iteration number of 5000, and 10,000 randomly generated pseudo-absences within the study area. The MaxEnt algorithm iterated until convergence, with a convergence threshold set at 0.00001. Model performance was independently evaluated using Receiver Operating Characteristic (ROC) curve analysis, and the Area Under the Curve (AUC) value was utilized as an indicator. The AUC categories were defined as excellent (AUC > 0.9), good ($0.8 \leq$ AUC < 0.9), moderate ($0.7 \leq$ AUC < 0.8), poor ($0.6 \leq$ AUC < 0.7), and fail (AUC < 0.6) [21]. Furthermore, the Jackknife Test was conducted to assess the contribution and importance of each environmental variable. The model output, in ASC format, was refined into raster layers using ArcGIS 10.8 for habitat suitability classification. Based on the suitability values derived from the simulation results, potential habitats were categorized into four classes: highly suitable habitat (suitability value > 0.6), moderately suitable habitat ($0.4 <$ suitability value $\leq 0.6$), low suitable habitat ($0.2 <$ suitability value $\leq 0.4$), and unsuitable habitat (suitability value $\leq 0.2$) [20,21].

To facilitate an in-depth analysis of the distribution patterns of the three Asian horseshoe crab species and accurately quantify the degree of niche overlap among them, the "*ecospat*" package was employed in R [30]. The Schoener index (D) was chosen as it is widely used in ecological niche studies, providing a robust measure of niche overlap by considering both presence differences and habitat suitability intensity. Schoener's D index, ranging from 0 (indicating no overlap) to 1 (indicating complete overlap), was used to quantify niche overlap. Additionally, niche similarity between each pair of

**Table 1.** Global marine environmental variable data (including physical variables, chemical variables, and nutrient variables) and the variance inflation factor (VIF) values of variables after screening through VIF analysis. Variables with VIF values greater than 10 were removed.

| Environment Variable | | *T. tridentatus* | *C. rotundicauda* | *T. gigas* | Grid Layer Code |
|---|---|---|---|---|---|
| **Physical** | Depth | **1.343** | **1.559** | **1.940** | gb_depth |
| | Slope | >10 | >10 | >10 | gb_slope |
| | Aspect (East-West) | >10 | >10 | >10 | msp_asp_ew |
| | Aspect (North-South) | >10 | >10 | >10 | msp_asp_ns |
| | Distance to Land | **1.576** | **1.834** | **1.695** | gb_land_distance |
| | Distance to Port | >10 | >10 | >10 | port_distance |
| | Ice Concentration Annual | >10 | >10 | >10 | aq_ice_amn |
| | Ice Concentration (May-Oct) | >10 | >10 | >10 | aq_ice_sum |
| | Ice Concentration (Nov-Apr) | >10 | >10 | >10 | aq_ice_win |
| | Tide average | **1.480** | **1.191** | **1.671** | kg_tide_average |
| | Wave height | **9.496** | **6.564** | **5.693** | aq_waveheight |
| | Wind speed | **2.835** | **3.694** | **1.418** | kg_wind_speed |
| | Surface Current | **3.280** | **1.846** | **1.311** | ecco2_uv_surface_current |
| | Euphotic Layer Bottom Depth | >10 | >10 | >10 | gc_zeu_mean |
| | Diffuse attenuation coefficient | >10 | >10 | >10 | bo_da_mean |
| | Maximum Temperature | **7.383** | **5.475** | **4.509** | bo_sst_maximum |
| | Average Temperature | >10 | >10 | **5.522** | bo_sst_average |
| | Minimum Temperature | >10 | >10 | >10 | bo_sst_minimum |
| | Range Temperature | >10 | >10 | >10 | bo_sst_range |
| | Summer Temperature (May-Oct) | **1.597** | **1.595** | **1.859** | na_may_oct_sst |
| | Winter Temperature (Nov-Apr) | >10 | >10 | >10 | na_nov_apr_sst |
| | Seabed Temperature | >10 | >10 | >10 | kg_b_temp |
| | Water Column Temperature | >10 | >10 | >10 | na_b_temp |
| | Salinity Mean | **5.859** | **6.064** | >10 | bo_salinity |
| | Bottom Salinity | >10 | >10 | >10 | na_b_salinity |
| | Photosynthetically Active Radiation | **8.304** | **4.183** | **5.918** | bo_parmean |
| **Chemical** | Average Chlorophyll-a | >10 | >10 | >10 | bo_chla_average |
| | Maximum Chlorophyll-a | >10 | >10 | >10 | bo_chla_maximum |
| | Minimum Chlorophyll-a | **2.648** | **3.637** | **2.594** | bo_chla_minimum |
| | Chlorophyll-a Range | **1.972** | **3.901** | **3.354** | bo_chla_range |
| | Summer (May-Oct) Maximum Chlorophyll-a | **1.875** | **2.997** | >10 | kg_chla_summax |
| | Winter (Nov-Apr) Maximum Chlorophyll-a | **1.821** | >10 | **1.523** | kg_chla_winmax |
| | Primary Productivity | **6.321** | **6.627** | >10 | aq_primprod |
| | pH | **3.888** | **6.599** | >10 | bo_ph |
| | Total Suspended Matter | >10 | >10 | >10 | gc_tsm_mean |
| **Nutrients** | Calcite | >10 | >10 | >10 | bo_calcite |
| | Nitrate | **7.558** | >10 | >10 | bo_nitrate |
| | Bottom Nitrate | >10 | >10 | >10 | kg_b_nitrate |
| | Phosphate | >10 | >10 | >10 | kg_phosphate |
| | Bottom Phosphate | >10 | >10 | >10 | kg_b_phosphate |
| | Silicate | **8.628** | **3.916** | **2.437** | bo_silicate |
| | Bottom Silicate | **3.857** | >10 | **2.488** | kg_b_silicate |
| | Dissolved Oxygen | **9.175** | **8.670** | **2.009** | bo_o2dis |
| | Bottom Dissolved Oxygen | **7.893** | **2.838** | **3.453** | kg_b_o2dissolve |
| | Saturated Oxygen | **7.529** | **1.475** | **3.203** | kg_s_o2saturate |

*(Continued)*

**Table 1.** (Continued)

| Environment Variable | | _T. tridentatus_ | _C. rotundicauda_ | _T. gigas_ | Grid Layer Code |
|---|---|---|---|---|---|
| | Bottom Utilized Oxygen | >10 | >10 | >10 | kg_b_o2utilized |
| | Particulate Organic Carbon | >10 | >10 | >10 | gc_poc_mean |
| | Particulate InOrganic Carbon | >10 | >10 | >10 | gc_pic_mean |

the three Asian horseshoe crab species was calculated using the "_ecospat_" package to understand and compare their positions and relationships within the ecological space [30]. The null hypothesis for niche similarity typically assumes that the degree of overlap between two species in niche space is random or solely determined by abiotic factors, without the influence of interspecific interactions.

## Results

### Assessment of model performance and identification of key environmental variables

After ROC curve analysis, the AUC values for the training and testing sets were above 0.99 for all species (_T. tridentatus_: 0.9963/0.9958, _C. rotundicauda_: 0.9980/0.9972, _T. gigas_: 0.9974/0.9972), indicating high model reliability (Table 2). Variance Inflation Factor (VIF) analysis was conducted on 48 marine environmental variables, which were categorized into 26 physical environmental factors, 9 chemical environmental factors, and 13 nutritional environmental factors. Following the VIF analysis, 21 environmental variables were retained for _T. tridentatus_, 19 for _C. rotundicauda_, and 21 for _T. gigas_ for modeling and analysis. The VIF values for these selected variables are presented in Table 1. The results of the MaxEnt model analysis revealed the environmental variables that most significantly shape the current potential habitats of the three Asian horseshoe crab species. These variables have crucial impacts on horseshoe crab distribution; for instance, water depth may be related to the availability of spawning substrate, while chlorophyll-a reflects primary productivity, influencing food availability. Wind velocity may affect larval dispersal and sediment transport. Specifically, for _T. tridentatus_, the variables in descending order of importance were: water depth (29.75%), minimum chlorophyll-a concentration (23.98%), distance to land (13.49%), wave height (8.59%), and silicate (7.05%). For _C. rotundicauda_, the variables were: minimum chlorophyll-a concentration (55.85%), distance to land (14.28%), summer temperature (May-Oct) (11.17%), water depth (10.03%), and silicate (2.12%). Lastly, for _T. gigas_, the variables were: water depth (50.71%), nitrate (11.7747%), wind speed (11.6947%), summer temperature (May-Oct) (8.1283%), and distance to land (6.5095%). Further evaluation of the influence of environmental variables on the potential habitat distribution of three species of Asian horseshoe crabs was conducted using a jackknife test, enabling the identification of the five most significant environmental factors for each species. For _T. tridentatus_, the order of importance of environmental factors affecting its potential habitat was as follows: water depth＞distance to land＞maximum chlorophyll-a content during summer＞minimum chlorophyll-a content＞bottom silicate (Fig 2A). For _C. rotundicauda_, the order of importance for its potential habitat distribution was: water depth＞distance to land＞minimum chlorophyll-a content＞maximum chlorophyll-a content during summer＞chlorophyll-a range (Fig 2B). Lastly, for _T. gigas_, the order of importance for its potential habitat distribution was: water depth＞distance to land＞wind speed＞maximum temperature＞maximum chlorophyll-a content (Fig 2C).

### Potential habitats of _T. tridentatus_, _C. rotundicauda_, and _T. gigas_

The potential habitat distribution maps for _T. tridentatus_, _C. rotundicauda_, and _T. gigas_, as predicted by the model, detail the regions of highly suitable, moderately suitable, and lowly suitable habitats, as illustrated specifically in Fig 3.

 _T. tridentatus_ exhibits a broad distribution range, spanning multiple regions of East Asia. Its northern boundary starts along the coastal areas of Kyushu Island in Japan and then extends south along the coastline of southern China, encompassing the western part of Taiwan Island and Hainan Island. It further traverses the northern section of the Tonkin Gulf

 

**Table 2. AUC values for three Asian horseshoe crabs based on Receiver Operating Characteristic (ROC) curve analysis.**

| Species | Training AUC | Testing AUC |
| --- | --- | --- |
| *T. tridentatus* | 0.9963 | 0.9958 |
| *C. rotundicauda* | 0.9980 | 0.9972 |
| *T. gigas* | 0.9974 | 0.9972 |

in Vietnam, reaching the Gulf of Thailand and continuing westward to the Andaman Sea. Its southern boundary stretches to the western coast of the Malay Peninsula and the island of Kalimantan. The core distribution areas of *T. tridentatus* are concentrated in the coastal regions of southern China and the Beibu Gulf, demonstrating the species' strong adaptability to these regions (see Fig 3A for details).

In contrast, the distribution range of *C. rotundicauda* is more southern-oriented. Its distribution starts on the western shore of the Taiwan Strait and extends north to cover the northern parts of the Tonkin Gulf and the coastal areas of the Gulf of Thailand. It further expands south, encompassing the coastal regions of the Malay Peninsula, Sumatra, and Kalimantan. The westernmost limit of its distribution reaches the northern coast of the Bay of Bengal. The core distribution areas of *C. rotundicauda* are located in the Gulf of Thailand, the coastal zones of the Malay Peninsula, and the coastal regions of Kalimantan, which are crucial for the survival of this species (Fig 3B).

Regarding *T. gigas*, its northern boundary in the South China Sea is roughly located off the coast of Nha Trang, Vietnam, and it can extend westward to the coastal waters along the Western Ghats of the Indian Peninsula. Overall, the distribution range of *T. gigas* is quite extensive, starting from the southern part of the South China Sea and extending along the Gulf of Thailand, the Malay Peninsula, Kalimantan, Sumatra, and reaching as far as Java Island. It even bypasses the Bay of Bengal westward, reaching the coastal waters along the Western Ghats of the Indian Peninsula. The core distribution areas are located along the Gulf of Thailand and the Malay Peninsula (Fig 3C).

### Key environmental factors influencing the potential habitats of *T. tridentatus*, *C. rotundicauda*, and *T. gigas*

For the three species of Asian horseshoe crabs, namely *T. tridentatus*, *C. rotundicauda*, and *T. gigas*, three of the most crucial environmental variables were selected for analysis as key environmental factors. Water depth, distance to land, and summer maximum chlorophyll-a concentration were identified as the pivotal factors influencing the potential habitat distribution of *T. tridentatus*. This species was found to inhabit waters with depths of up to 80 meters, within approximately 500 meters from the shore, and where the summer maximum chlorophyll-a concentration did not exceed 20 µg/L. The primary distribution was concentrated in waters within 0–200 meters from the shore and at depths of up to 40 meters (Fig 4A–C). For *C. rotundicauda*, water depth, distance to land, and minimum chlorophyll-a concentration emerged as the key factors shaping its potential habitat distribution. This species was distributed in waters with depths of up to 30 meters, within approximately 160 meters from the shore, and where the minimum chlorophyll-a concentration did not exceed 9 µg/L. The main distribution was focused in waters within 50–120 meters from the shore, at depths of up to 10 meters, and with minimum chlorophyll-a concentrations of 5 µg/L or less (Fig 4D–F). As for *T. gigas*, water depth, distance to land, and wind speed were determined to be the critical factors influencing its potential habitat distribution. This species was found in waters with depths of up to 70 meters, within approximately 500 meters from the shore, and where wind speeds did not exceed 5 m/s. The primary distribution was centered in waters within 50–200 meters from the shore, at depths of up to 40 meters, and with wind speeds of 1.5 m/s or less (Fig 4G–I).

### Analysis of niche overlap and habitat similarity among three species of Asian horseshoe crabs

The niche overlap coefficient, Schoener's D, between *T. tridentatus* and *C. rotundicauda* is 0.3203158, indicating a certain degree of overlap in ecological resources such as food and habitat, though not to a high degree of consistency (Fig 5A).

                                                     

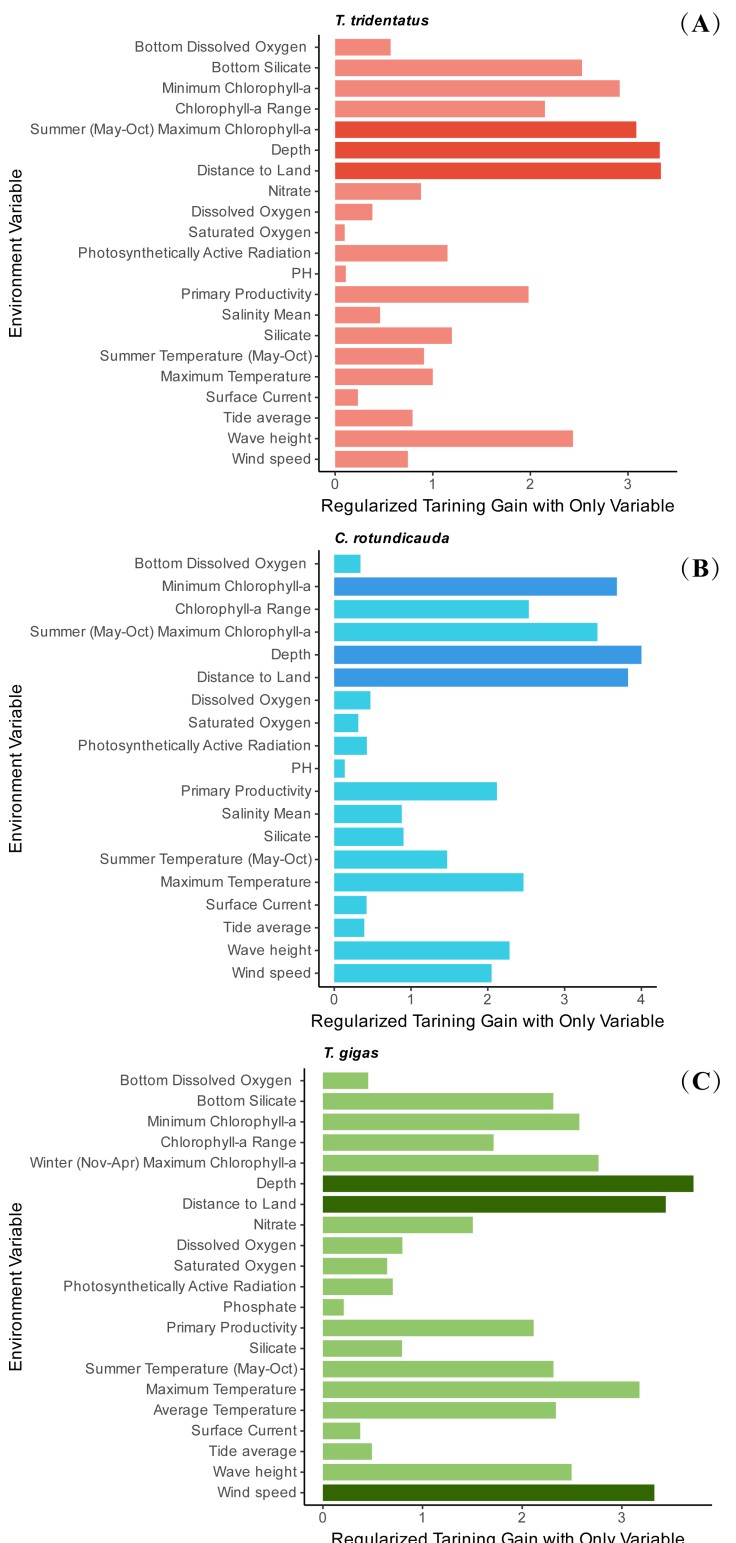

**Fig 2. Importance analysis of environmental variables for the habitats of _T. tridentatus_ (A), _C. rotundicauda_ (B), and _T. gigas_ (C) based on Jackknife Test.**

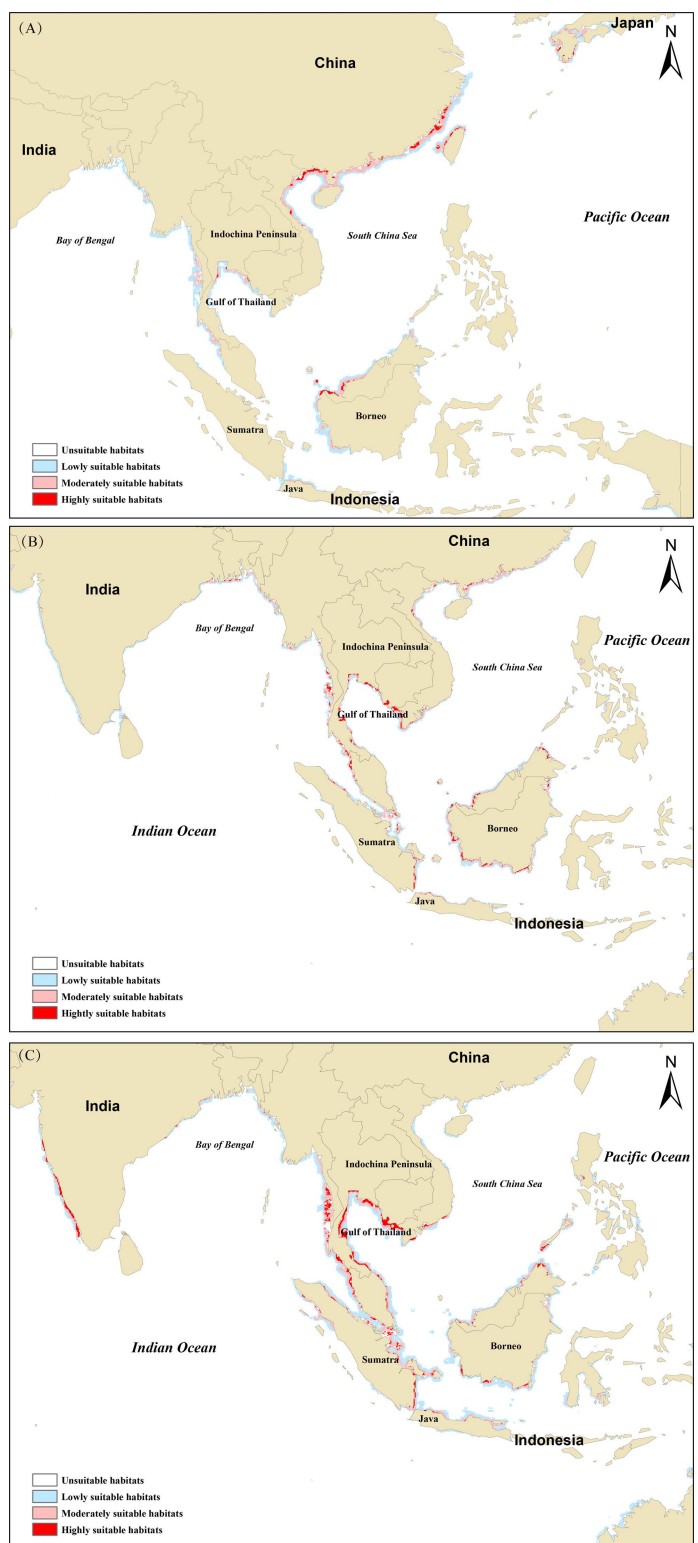

**Fig 3. Distribution of potential habitats with different suitability levels for _T. tridentatus_ (A), _C. rotundicauda_ (B), and _T. gigas_ (C).**

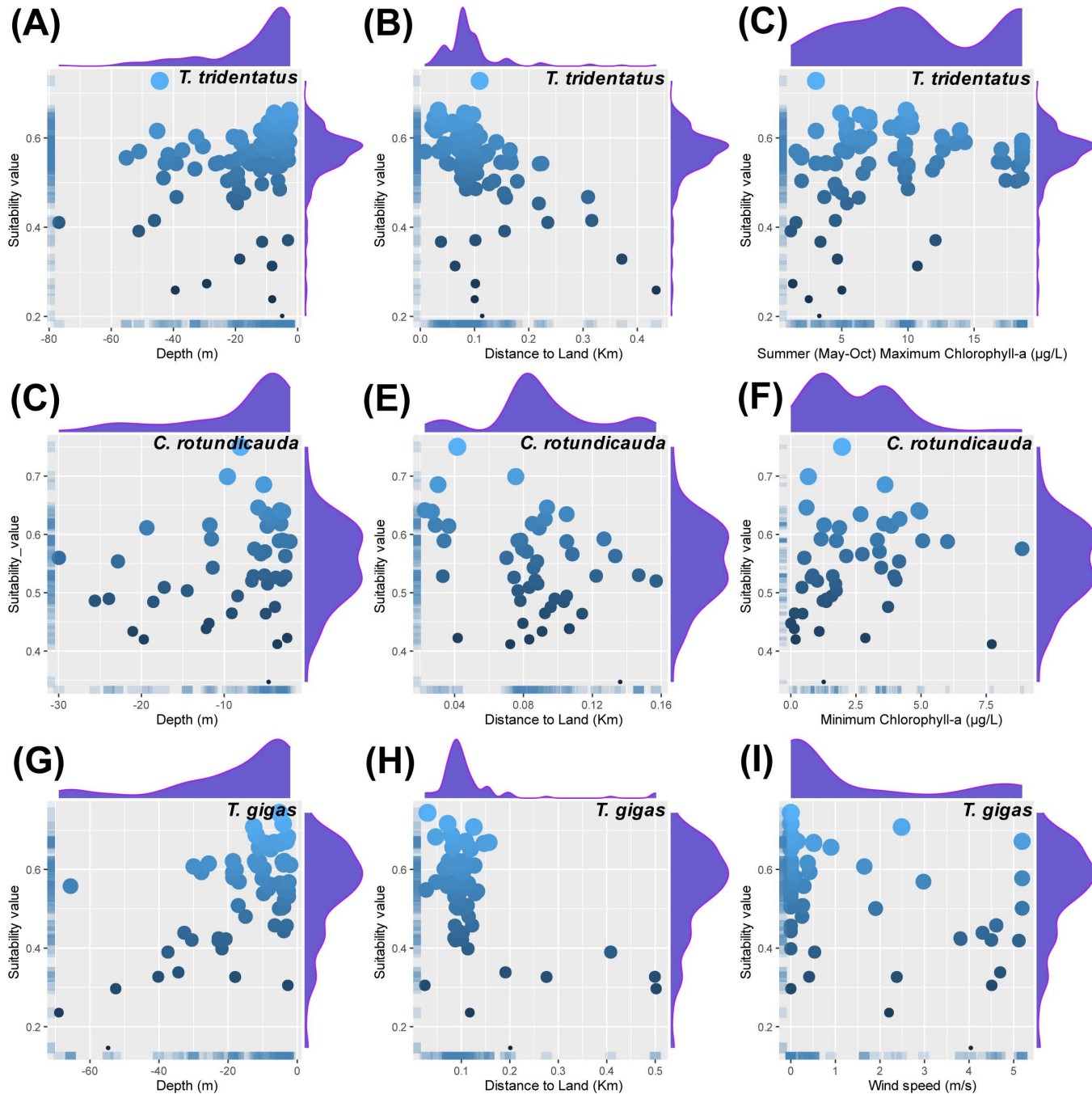

**Fig 4. Scatter plots of key environmental factors versus suitability values for the potential habitats of *T. tridentatus*, *C. rotundicauda*, and *T. gigas*.** (A) Depth vs. suitability value for *T. tridentatus*, (B) Distance to land vs. suitability value for *T. tridentatus*, (C) Maximum chlorophyll-a in summer vs. suitability value for *T. tridentatus*, (D) Depth vs. suitability value for *C. rotundicauda*, (E) Distance to land vs. suitability value for *C. rotundicauda*, (F) Minimum chlorophyll-a vs. suitability value for *C. rotundicauda*, (G) Depth vs. suitability value for *T. gigas*, (H) Distance to land vs. suitability value for *T. gigas*, (I) Wind speed vs. suitability value for *T. gigas*.

**Fig 5. Niche Overlap and Similarity *T. tridentatus*, *C. rotundicauda*, and *T. gigas*.** (A) Niche Overlap between *T. tridentatus* and *C. rotundicauda*, (B) Niche Similarity between *T. tridentatus* and *C. rotundicauda*, (C) Niche Overlap between *T. tridentatus* and *T. gigas*, (D) Niche Similarity between *T. tridentatus* and *T. gigas*, (E) Niche Overlap between *C. rotundicauda* and *T. gigas*, (F) Niche Similarity between *C. rotundicauda* and *T. gigas*.

This overlap may stem from their similar ecological needs and adaptation strategies, such as comparable positions in the food chain and similar habitat preferences. However, the habitat similarity p-value of 0.02398 suggests that, while there are commonalities in their habitats, significant differences also exist, which contribute to their coexistence and differentiation within the same ecosystem (Fig 5B). Additionally, the niche overlap coefficient between *T. tridentatus* and *T. gigas* is lower (Schoener's D = 0.2200121), and the habitat similarity p-value is even smaller (p = 0.00999), indicating more pronounced differences in their ecological needs and habitats (Fig 5C, D). These differences may arise from different adaptation strategies formed during evolution, such as differentiation in food choice and specialization of habitats, which help reduce interspecific competition and promote species coexistence. Conversely, the niche overlap coefficient between *C. rotundicauda* and *T. gigas* is higher (Schoener's D = 0.6221374), and the habitat similarity p-value is relatively low (p = 0.01399), indicating a higher similarity in ecological resources and habitats (Fig 5E, F). This high degree of niche overlap and habitat similarity may increase interspecific competition pressure between them, but it may also prompt them to develop more refined differentiation strategies in ecological adaptation and evolution, such as temporal niche partitioning and fine-grained division of food resources, to maintain population stability and ecological balance. The low overlap between *T. tridentatus* and *T. gigas* suggests ecological segregation, possibly due to distinct habitat and dietary preferences. In contrast, the high overlap between *C. rotundicauda* and *T. gigas* may indicate stronger competition for resources, requiring additional niche differentiation mechanisms. *T. tridentatus*, *C. rotundicauda*, and *T. gigas* exhibit different characteristics in niche overlap and habitat similarity, which not only reflect their coexistence mechanisms within the same ecosystem but also provide important clues for deepening the understanding of interspecies interactions and evolutionary relationships.

## Discussion

### The selection of the model and the accuracy of the results

Species distribution models integrate species occurrence data with environmental variables to estimate species' ecological niches through specific algorithms, projecting these onto geographic space to reveal species presence probability, habitat suitability, or species richness [19]. In this study, MaxEnt was selected to simulate potential habitats for three Asian horseshoe crab species, representing one of the most extensively applied SDMs. The theoretical underpinning of MaxEnt originates from the second law of thermodynamics, which posits that a non-equilibrium living system sustains its existence via energy exchange with the environment, demonstrating "dissipative" characteristics. As the system's entropy progressively increases and approaches its maximum value, the living system attains a dynamic equilibrium with the external environment. Within the realm of species potential distribution research, species and their habitats can be regarded as a coupled system. By calculating characteristic parameters of the system under maximum entropy conditions, stable correlations between species and their environments can be ascertained, facilitating predictions of species distribution. Based on this principle, the MaxEnt model constructs a maximum entropy probability distribution function to accurately estimate species' potential distributions, conditioned on known species occurrence locations and environmental variables [19]. Beyond MaxEnt, early habitat prediction models also encompassed BIOCLIM, HABITAT, DOMAIN, ENFA, Generalized Linear Models (GLMs), and Generalized Additive Models (GAMs) [19,31–37]. Elith et al. (2006) conducted a comparative model evaluation utilizing distribution data of vascular plants, birds, reptiles, and mammals across six global regions (Australian Wet Tropics, New South Wales, Ontario, New Zealand, South America, and Switzerland). Their findings demonstrated that MaxEnt exhibited the best performance, followed by GLMs and GAMs, with HABITAT and DOMAIN models showing moderate performance, and BIOCLIM yielding the poorest predictive outcomes [38]. Subsequent studies have further substantiated the advantages of MaxEnt in terms of predictive accuracy and robustness [20,21,39]. In this study, the results of Asian horseshoe crab habitat modeling were evaluated using ROC curves, with AUC values basically above 0.9, confirming the reliability of model selection and the precision of results. This finding is also consistent with the experiences of many researchers engaged in horseshoe crab resource surveys and conservation efforts [4,40–42].

**Key environmental factors influencing the distribution of three species of Asian horseshoe crabs**

Water depth and distance to land are the most critical factors affecting the distribution of the three Asian horseshoe crabs, together shaping their distribution pattern. For different Asian horseshoe crab species, summer maximum chlorophyll-a, minimum chlorophyll-a concentration and wind speed affect their niche differences. Water depth is one of the significant factors influencing the distribution of marine habitats [43,44]. For Asian horseshoe crabs, variations in water depth can potentially affect their survival and reproduction [45]. Different species of Asian horseshoe crabs may adapt to distinct water depth ranges. For instance, our research reveals that *C. rotundicauda* may prefer to inhabit shallower waters within 10 meters, whereas *T. tridentatus* and *T. gigas* may be more adapted to deeper environments within 40 meters. Changes in water depth can lead to alterations in the living space of horseshoe crabs, subsequently impacting their population sizes and distributions [10]. Furthermore, variations in water depth may also influence the food sources of horseshoe crabs [40,46]. Water depth affects the distribution and abundance of horseshoe crab food sources such as plankton and benthic organisms. For example, in shallow water areas with sufficient light, the growth of phytoplankton is promoted, which in turn provides food sources for benthic organisms and indirectly affects the food supply of horseshoe crabs. Additionally, horseshoe crabs migrate annually in response to seasonal and water temperature changes. Every November, they swim from shallow seas to deep-water areas for overwintering; the following April to May, they return to shallow seas for reproductive migration. During migration, horseshoe crabs move between regions with different water depths and obtain different food sources. For example, while overwintering in deep-water areas, they may feed on certain deep-sea organisms; after returning to shallow seas, their food sources change again. In deeper waters, horseshoe crabs may need to seek out more food sources to sustain their survival [46].

The distance from the shore (distance to land) is also a crucial factor influencing the distribution of marine habitat, and variations in this distance can potentially impact the migration and reproduction patterns of marine organisms. The Asian horseshoe crab, for instance, may select different migration routes based on changes in the distance from the shore. Waters closer to the shore are more susceptible to human activities, such as pollution and fishing, which may compel horseshoe crabs to migrate to more distant waters. Furthermore, the distance from the shore can also affect the reproductive environment of horseshoe crabs.

During their life cycles, *T. tridentatus*, *C. rotundicauda*, and *T. gigas* migrate in pairs to the vicinity of the high-tide zone in the intertidal zone to spawn[47]. The larvae of the horseshoe crab settle in nearshore muddy beaches, feeding on benthic organisms that are closely related to primary productivity, which is represented by chlorophyll-a. Consequently, chlorophyll-a content is actually a very important environmental factor that influences the distribution of horseshoe crabs, particularly the minimum chlorophyll-a concentration and the maximum chlorophyll-a concentration during the summer breeding season. Wind exerts a profound influence on the survival of marine benthic organisms. Notably, short-term extreme storms can readily induce stranding of marine benthic organisms [48]. For instance, in the aftermath of Storm Emma (February 28 to March 5, 2018), a large-scale stranding event involving various benthic organisms occurred on Punta Umbria Beach in Spain [48]. Following long-term adaptive evolution, wind speed has come to shape the composition and distribution patterns of benthic communities. For the Asian horseshoe crab, specifically *T. gigas*, variations in wind speed primarily affect its activity range and survival condition. Strong winds may force the horseshoe crabs to migrate to safer waters, whereas weaker winds may expand their activity range. Additionally, wind speed can potentially influence the reproduction success rate of the horseshoe crabs. During the breeding season, strong winds may disrupt the reproductive environment of the horseshoe crabs, thereby reducing their reproduction success rate. In summary, depth influences not only the availability of suitable spawning substrate but also access to food sources and protection from predators. Similarly, chlorophyll-a reflects primary productivity, impacting the abundance of organisms forming the horseshoe crab food base.

Although temperature did not rank among the top three key factors influencing the distribution of Asian horseshoe crabs, its ecological significance remains non-negligible. Importance ranking analysis demonstrated that maximum

temperature and summer water temperature (May-Oct) held a certain weight in the model (Fig 2), a result closely linked to the tropical-subtropical adaptability of Asian horseshoe crabs, particularly for the mangrove horseshoe crab *C. rotundicauda* and coastal horseshoe crab *T. gigas* near the equator. However, for the relatively north-distributed Chinese horseshoe crab (*T. tridentatus*), its population exhibits a migratory strategy of "deepwater overwintering-shallow coastal breeding": migrating to 60-meter deep waters in winter to avoid low-temperature stress and returning to nearshore areas for reproduction as water temperatures rise in summer [40,49]. This behavioral pattern underscores the pivotal role of temperature in shaping their survival strategy—deepwater zones, characterized by high thermal mass and sufficient vertical mixing, create a relatively stable environment (winter temperature fluctuations in deep-sea areas are far smaller than those in shallow waters), providing overwintering refugia for horseshoe crabs. Conversely, rising summer temperatures in shallow coastal zones directly trigger reproductive behaviors. From a spatial dimension, offshore distance and water depth further amplify temperature's influence on horseshoe crab distribution through thermal regulation. Offshore deepwater zones experience smaller temperature fluctuations due to thermal buffering effects, which may partially explain why temperature did not emerge as the most critical limiting factor (other factors such as water depth already partially reflect temperature effects). Nevertheless, under global warming scenarios, rising seawater temperatures and increased frequency of extreme climate events may disrupt existing temperature-depth-offshore distance equilibria [50]. For example, if deep-sea winter temperatures rise significantly due to climate warming, this could alter the overwintering migratory patterns of Chinese horseshoe crabs. Similarly, abnormally high summer temperatures may exceed the physiological tolerance thresholds of horseshoe crabs, leading to reduced reproductive success rates. Therefore, future research should employ more refined temperature response experiments (e.g., critical thermal tolerance tests, studies on embryonic development under temperature fluctuations) to quantify the physiological vulnerability of horseshoe crabs and establish distribution prediction frameworks coupled with climate models. Such efforts will provide scientific bases for formulating marine protected area plans based on thermal adaptation thresholds and optimizing conservation strategies for migratory corridors, especially crucial for mitigating thermal stress in tropical-subtropical marine ecosystems under global warming pressures.

## Niche overlap and resource utilization of three Asian horseshoe crabs

This study analyzed the niche overlap and habitat similarity among three species of Asian horseshoe crabs, revealing complex relationships in their resource utilization and habitat selection, and providing insights into their coexistence mechanisms. The niche overlap coefficient between *T. tridentatus* and *C. rotundicauda* indicated a certain degree of overlap in resources such as food and habitat, but without high consistency. This moderate overlap may reflect their similar ecological needs and adaptation strategies [51], such as comparable positions in the food chain and similar habitat preferences [44]. However, the p-value for habitat similarity showed that, despite commonalities, there were still significant differences in habitat selection between the two species. These differences may contribute to their coexistence in the same ecosystem and reduce direct competition through fine differentiation of niches. In contrast, the niche overlap coefficient between *T. tridentatus* and *T. gigas* was lower, and the p-value for habitat similarity was even smaller, indicating more significant differences in their ecological needs and habitats [52]. These differences may stem from different adaptation strategies formed during evolution, such as differentiation in food choice and specialization of habitats. These strategies not only help reduce interspecific competition but may also promote species coexistence and diversity [53]. Notably, the niche overlap coefficient between *C. rotundicauda* and *T. gigas* was higher, while the p-value for habitat similarity was relatively low. This high degree of niche overlap and habitat similarity may increase interspecific competition pressure between them. However, it may also prompt them to develop more refined differentiation strategies during ecological adaptation and evolution, such as temporal niche partitioning and fine-grained division of food resources. These strategies are crucial for maintaining population stability and ecological balance.

## Conclusions

This study offers a comprehensive analysis of the habitat characteristics, current potential habitat distributions with varying suitability levels, and niche overlap and similarity among three Asian horseshoe crab species: *T. tridentatus*, *C. rotundicauda*, and *T. gigas*. The findings reveal that water depth and distance to land are shared critical factors shaping the potential habitat distribution of these species. However, each species is also influenced by distinct environmental variables: *T. tridentatus*'s distribution is further limited by summer (May-Oct) maximum chlorophyll-a concentration, *C. rotundicauda*'s habitat is closely tied to minimum chlorophyll-a concentration, and *T. gigas*'s habitat selection is notably impacted by wind speed. In terms of niche overlap, *C. rotundicauda* and *T. gigas* show the highest overlap (Schoener's D = 0.62), signifying similar ecological resource use, whereas *T. tridentatus* and *T. gigas* exhibit the lowest overlap (Schoener's D = 0.22), indicating clear niche differentiation. This research provides crucial insights for the conservation and management of these evolutionarily important "living fossils." To enhance protection, we recommend establishing marine protected areas (MPAs) in identified critical habitats, particularly focusing on regions where habitat overlap is high, to safeguard shared ecological resources. Additionally, strengthening regulations on the harvest of these species for biomedical purposes is essential to balance conservation with sustainable use. Proactive strategies are needed to assess the impacts of global warming and ocean acidification on horseshoe crab distributions. This includes monitoring potential habitat shifts to new areas due to changing environmental conditions, such as alterations in chlorophyll-a concentrations and wind speed, to inform adaptive conservation measures. The study underscores the value of MaxEnt modeling as a predictive tool in conservation. By identifying habitat distribution patterns and offering critical insights through predictive data, MaxEnt modeling advances our ecological understanding of these species, aiding in the development of effective conservation strategies. Future applications of this model could simulate habitat changes under different climate scenarios, further supporting adaptive management decisions.

## Supporting information

**S1 File. *Tachypleus tridentatus*.**
(XLS)

**S2 File. *Carcinoscorpius rotundicauda*.**
(XLS)

**S3 File. *Tachypleus gigas*.**
(XLS)

## Author contributions

**Conceptualization:** Jian Liao, Zhong-Duo Wang.

**Data curation:** Jian Liao, Shui-Yuan Zhang, Zhong-Duo Wang.

**Formal analysis:** Jian Liao.

**Funding acquisition:** Jian Liao.

**Investigation:** Jian Liao, Shui-Yuan Zhang, Yu-Song Guo, Zhong-Duo Wang.

**Methodology:** Jian Liao, Yu-Song Guo.

**Software:** Jian Liao.

**Supervision:** Zhong-Duo Wang.

**Writing – original draft:** Jian Liao.

**Writing – review & editing:** Jian Liao, Chun-Hui Xiong, Gao-Cong Li, Jia-Yu Li, Yuan-Feng Yang, Shui-Yuan Zhang, Yi-Yang Li, Kai-Lin Zeng, Mei-Ling Hu, Zhong-Duo Wang.

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
