## [Decision Letter · Decision Letter 0]

25 Mar 2025

PONE-D-25-08646Exploring habitat characteristics, distribution, and niche relationships of Asian horseshoe crabs: preserving living fossils and biodiversityPLOS ONE

Dear Dr. Wang,

Thank you for submitting your manuscript to PLOS ONE. After careful consideration, we feel that it has merit but does not fully meet PLOS ONE’s publication criteria as it currently stands. Therefore, we invite you to submit a revised version of the manuscript that addresses the points raised during the review process.

We look forward to receiving your revised manuscript.

Kind regards,

Phuping Sucharitakul

Academic Editor

PLOS ONE

Journal Requirements:

2**. ** Thank you for stating the following financial disclosure:

“This study was funded by the Guangdong Provincial Field Observation and Research Station for Marine Ecosystem in Hanjiang River Estuary - Nanao Island Area Open Fund (Grant No. HNS202407), the National Key Research and Development Program of China (Grant No. 2024YFD2401803 & 2024YFD2401802), the Guangdong Provincial Ordinary University Youth Innovative Talent Project in 2024 (Grant No. 2024KQNCX134), the Guangdong Provincial Special Fund Project for Talent Development Strategy in 2024 (Grant No. 2024R3005), and the Guangdong Ocean University Scientific Research Startup Funding Project (Grant No. 060302022312).”

Reviewers' comments:

Reviewer's Responses to Questions

**Comments to the Author**

1. Is the manuscript technically sound, and do the data support the conclusions?

Reviewer #1: Partly

Reviewer #2: Yes

Reviewer #3: Yes

2. Has the statistical analysis been performed appropriately and rigorously? 

Reviewer #1: Yes

Reviewer #2: Yes

Reviewer #3: Yes

3. Have the authors made all data underlying the findings in their manuscript fully available?

Reviewer #1: Yes

Reviewer #2: Yes

Reviewer #3: Yes

4. Is the manuscript presented in an intelligible fashion and written in standard English?

Reviewer #1: Yes

Reviewer #2: No

Reviewer #3: Yes

5. Review Comments to the Author

Reviewer #1: The manuscript presents a valuable analysis of the distribution and habitat characteristics of three Asian horseshoe crab species using MaxEnt modeling and niche analysis. The use of MaxEnt modeling is appropriate for species distribution predictions, and the study follows standard procedures for model validation (e.g., AUC assessment, VIF analysis to remove collinearity).

The incorporation of paleo-environmental modeling to assess Last Glacial Maximum (LGM) refugia is a valuable approach for understanding historical species distributions.

The study has strong potential for contributing to conservation efforts by identifying key environmental factors affecting species distribution. However, there are major issues with how the findings are framed, interpreted, and connected to the broader context of conservation and biodiversity management. The manuscript would benefit from a stronger positioning within the state of the art, improved discussion of methodological assumptions, and clearer conservation implications. The Introduction does not sufficiently compare the study’s approach and findings with prior research on horseshoe crab distribution. The manuscript should clearly specify:

--What aspects of habitat suitability and niche relationships have already been studied for these species.

What specific gaps this study addresses (e.g., finer-scale niche overlap analysis, previously overlooked environmental factors).

-How the findings advance current understanding.

The Discussion and Conclusion sections claim that the study supports conservation strategies but fail to articulate:

How the identified key environmental factors should inform conservation efforts.

Specific recommendations for habitat protection or management actions.

Potential policy implications (e.g., marine protected areas, restoration initiatives).

The manuscript does not clarify whether occurrence data correspond to adults, juveniles, or larvae, which is critical for interpreting distribution patterns. Horseshoe crabs exhibit ontogenetic habitat shifts, with larvae often requiring different environmental conditions than adults.

Specific comments are found in the attached pdf and must be addressed individually.

Reviewer #2: The study presents a robust analysis of the distribution and habitat characteristics of Asian horseshoe crabs (Tachypleus tridentatus, Tachypleus gigas, and Carcinoscorpius rotundicauda), employing predictive modeling to infer both historical and contemporary patterns. The use of the MaxEnt model and niche overlap metrics are well-established and appropriate methodological approaches for the study's objectives.

Despite its scientific relevance and potential impact on species conservation, the manuscript has some weaknesses, including: 1) a lack of deeper justifications for certain methodological choices; 2) some model-based inferences that require stronger support from the literature; and 3) minor textual adjustments to improve clarity and argumentation.

More information about suggestions the authors can to access in the PDF archive annex to this review.

Thus, I recommend that the manuscript undergo revision before acceptance for publication, incorporating the suggested adjustments outlined below.

Reviewer #3: Overall:

This is overall a very good and interesting paper. The authors used publicly available datasets to model the spatial distribution, habitat suitability, and niche overlap among the three Asian species of horseshoe crabs, with the aim of identifying important habitats for conservation planning for these threatened or potentially threatened (data-deficient, understudied) species. Exercises were also done to identify the key environmental variables determining each species’ habitat distribution and their potential habitat and refugia during the Last Glacial Maximum (LGM). For the most part, the paper is well written, the figures are good, and the analyses are done and interpreted logically. I particularly liked how the authors went through the key variables determining habitat that they identified in the Discussion to assess why these should be important for the studied species. The study is significant, novel, and should be published after some revisions. I make some comments below about places where the paper might benefit from clarifications or additional Discussion points, but with one exception these are mostly minor concerns. My one larger concern relates to the LGM analysis, which (maybe more by brevity/omission of details than intended misdirection) may appear to the reader as an equivalent analysis to the main modelling efforts, when in reality it was a much more limited exercise with less certain results. As I elaborate below, there needs to be much more clarity on exactly how this was done, and probably also more caution in interpreting its results.

Please note: I consider that Minor Revisions are needed for the majority of the paper, although suggested changes to the details presented for the LGM analysis (or even its removal) as detailed below may be considered more extensive (Major).

Specific comments, roughly in order of their occurrence in the text:

(1) This is a very minor gripe, but it confused that the three horseshoe crab species’ names are listed in no apparent order for no clear reason in some parts of the paper (e.g., lines 129-130, 133-134, 215-216, 24-246). In other parts Tachypleus species are listed before C. rotundicauda, which at least has some logic (e.g., lines 55-6, 61-68, 85-86), although it is still a little strange to list T. tridentatus before T. gigas; this latter order does at least match the alphabetical order of the common names given in lines 54-56. I think it would be preferable to list the species’ names in actual alphabetical order at most/all places in the text – so Carcinoscorpius rotundicauda, Tachypleus gigas, and T. tridentatus. This would also apply to how the results are presented.

(2) Lines 90-104, 149-153: MaxENT was used for species distribution modelling in this study. Nothing wrong with this, it is a well-established and widely used method. The authors do note around line 90 that other techniques are available, however, and it may be worth noting alternative approaches in the Discussion somewhere. Could this analysis be repeated (maybe in future) with different approaches, such as what, with what added benefits/drawbacks, and would the outcomes be expected to be very different, etc.?

(3) Line 137: Just for clarity, should specify that this is “a resolution of 2.5 minutes latitude/longitude” (to avoid confusion with time).

(4) LGM Analysis: Lines 137-138: I think a lot of important detail is skipped over here.

The GMED database does include data layers for the Last Glacial Maximum (LGM), but as far as I can tell from browsing the website only a handful (4) of environmental variables are available there from the LGM – specifically water depth, sea surface temperature, salinity, and ice thickness. My understanding based on the paper is that the authors applied their habitat suitability/species distribution models for each of the three species derived for the present/historical period to the LGM data to determine past habitats and refugia (lines 293-315 in the Results, Fig. 6). However, key variables identified in the present-day models, such as chlorophyll-a and winds (lines 247-265), are not apparently available in the LGM data layers from GMED (indeed, of all the variables listed in Table 1, hardly any seem available in the LGM layers), so it is not clear to me how the authors could identify past habitats in this setup. This needs to be explained.

Maybe additional variables are available in GMED for the LGM that are not obvious on the public-facing webpage, but if so more information needs to be provided on where/how these were downloaded. A table like Table 1 showing which LGM data layers were considered would be useful regardless.

However, I suspect that maybe the present-day potential distributions were just applied to water depths and shorelines available during the LGM to identify past refugia, without considering other variables’ past values’ impact in habitat suitability. If so, not stating this clearly in the paper potentially misleads the reader. Further, there are important limitations to this approach that need to be addressed in the Methods and Discussion. For example, if chlorophyll concentration strongly impacts a species’ distribution, then can we be sure that an area with suitable chlorophyll levels today would have had suitable levels in the LGM, even if it was available habitat based on depth and shoreline? I think not. Consider that today’s mid-deeper water shelf seas would have been shallow coastal areas during the LGM, with correspondingly different currents, light levels, etc. impacting plankton productivity.

In the discussion using paleoenvironmental data (line 444) to better address this in the future is mentioned – is this because such data were not used in the present study? If so this is not obvious from the Methods and Results as written.

It might even be worth considering removing the LGM analysis entirely, given the potential problems identified above. I see it as a minor side analysis to the main part of this study, and do not think a lot of important material is lost by not identifying LGM refugia – the study still produces habitat suitability and niche overlap maps for the species and identifies important variables, which is most useful for conservation planning.

(5) Line 298: change “highly and moderately suitability” to “highly and moderately suitable”

(6) Lines 245-265, 338, and elsewhere: The authors should consider not presenting depths as negative numbers in the paper. It is logical and probably necessary for depth have a negative value when running your models (particularly to make it apply to the LGM when depths might become positive die to lower sea levels), but it become confusing to read in the text – most of us are used to thinking of the sea at a given point being 40 m deep, not “-40 m deep”. Some readers may even misread the negative values to imply depths above the sea level, below the sea bed, at some point in between the surface and bottom (e.g., 40 m down), etc.

(7) Temperature is included in the modeling exercise as an environmental variable, but is not one of the main ones found to impact the habitat distributions of the studied species. This is somewhat surprising, given the widespread impacts of temperature on biota, particularly aquatic arthropods, but may well be true – the studied species are all found in and around the tropics mainly, so temperature may have less impact on their biology overall within their ranges, other than perhaps at the extremes. This still may be a point to raise in the Discussion, though, and can be tied in with potential conservation concerns related to climate change. Also, during the LGM, one wonders if temperatures might have been different enough to affect habitat suitability for these species? Maybe another Discussion point to add. SST data are available for the LGM from GMED, so there is potential to address this question in a future exercise.

6. PLOS authors have the option to publish the peer review history of their article (what does this mean? ). If published, this will include your full peer review and any attached files.

**Do you want your identity to be public for this peer review?** For information about this choice, including consent withdrawal, please see our Privacy Policy .

Reviewer #1: No

Reviewer #2: **Yes: ** Marcelo Antonio Amaro Pinheiro

Reviewer #3: **Yes: ** Brady K. Quinn

---

## [Author Response · Author response to Decision Letter 1]

7 Apr 2025

Reviewer #1: The manuscript presents a valuable analysis of the distribution and habitat characteristics of three Asian horseshoe crab species using MaxEnt modeling and niche analysis. The use of MaxEnt modeling is appropriate for species distribution predictions, and the study follows standard procedures for model validation (e.g., AUC assessment, VIF analysis to remove collinearity).

The incorporation of paleo-environmental modeling to assess Last Glacial Maximum (LGM) refugia is a valuable approach for understanding historical species distributions.

The study has strong potential for contributing to conservation efforts by identifying key environmental factors affecting species distribution. However, there are major issues with how the findings are framed, interpreted, and connected to the broader context of conservation and biodiversity management. The manuscript would benefit from a stronger positioning within the state of the art, improved discussion of methodological assumptions, and clearer conservation implications. The Introduction does not sufficiently compare the study’s approach and findings with prior research on horseshoe crab distribution. The manuscript should clearly specify:

--What aspects of habitat suitability and niche relationships have already been studied for these species.

What specific gaps this study addresses (e.g., finer-scale niche overlap analysis, previously overlooked environmental factors).

-How the findings advance current understanding.

Response Thank you for your attentive comments. We have added a short paragraph at the end of the third section in the introduction, elaborating on the research of marine organisms, especially horseshoe crab habitat suitability and ecological niches, and pointing out that the environmental variables they used were climatic variables, which have a weak correlation with horseshoe crabs, a marine benthic organism. We have comprehensively considered marine environmental factors with greater impact, including physical, chemical, and nutritional variables, offering predictable advantages (Line 122-132). The specific added content is as follows:

For example, the MaxEnt model has been used to simulate scad fish (Decanters spp.) in the northern South China Sea [26], and joint species distribution models have been employed to predict the distribution of seafloor taxa and identify vulnerable marine ecosystems in New Zealand waters [27]. Regarding research on horseshoe crab habitat suitability and suitability relationships, Chinese scholars have conducted local studies in the Beibu Gulf region [28]. However, the environmental factors studied were limited to climatic factors, which have a weak correlation with horseshoe crabs, a marine benthic organism [28]. Our study, starting from the marine environment, comprehensively considers physical, chemical, and nutritional factors, which are more in line with the habitat conditions of horseshoe crabs, a marine benthic organism, and offers predictable advantages.

The Discussion and Conclusion sections claim that the study supports conservation strategies but fail to articulate:

How the identified key environmental factors should inform conservation efforts.

Specific recommendations for habitat protection or management actions.

Potential policy implications (e.g., marine protected areas, restoration initiatives).

Response Thank you for your valuable comments. In response to your feedback, we have made targeted additions to the Discussion and Conclusion sections to address the points raised (). Specifically, we have proposed the following conservation and regulatory measures:

1. Establishment of Marine Protected Areas (MPAs): We recommend the creation of MPAs in critical habitats, with a particular focus on areas with high habitat overlap, to protect shared ecological resources. This measure aims to safeguard these vital ecosystems and the species that depend on them.

2. Strengthening Regulations on Harvesting: We emphasize the need for stricter regulations on the harvest of horseshoe crabs for biomedical purposes. This is crucial to balance conservation efforts with sustainable use, ensuring that these species are not overexploited for short-term gains.

3. Proactive Strategies for Climate Change Impacts: Recognizing the potential impacts of global warming and ocean acidification on horseshoe crab distributions, we propose proactive strategies to assess these effects. This includes monitoring potential habitat shifts to new areas due to changing environmental conditions, such as alterations in chlorophyll-a concentrations and wind speed. By doing so, we can inform adaptive conservation measures and ensure that protection efforts remain effective in the face of changing environmental conditions.

These additions provide a comprehensive framework for conservation and management, addressing the need for specific recommendations and policy implications. We believe these measures will contribute to the long-term survival and sustainability of horseshoe crab populations and their ecosystems.

The manuscript does not clarify whether occurrence data correspond to adults, juveniles, or larvae, which is critical for interpreting distribution patterns. Horseshoe crabs exhibit ontogenetic habitat shifts, with larvae often requiring different environmental conditions than adults.

Response Thank you for pointing out this important issue. We used distribution data from adults, which are more accessible. Adult horseshoe crabs exhibit more stable habitat characteristics and more pronounced activity patterns, making their distribution data more systematically acquirable through field surveys, satellite tracking, and other means. In existing databases and literature records, the spatial and temporal continuity of adult distribution records is also more complete. We have added supplementary explanations in the corresponding sections, such as in the "Species occurrence sites" subsection within the Materials and Methods section (Line 149).

Specific comments are found in the attached pdf and must be addressed individually.

The title does not clearly reflect the study contents; there is no attempt to addressing results with species and biodiversity preservation. instead focus on the aims novelty in relation to state of the art

Response Thank you for your constructive feedback. We have carefully considered your comments in conjunction with those from another reviewer, and have revised the manuscript accordingly. In response to your specific concern regarding the title, we have modified it to better reflect the study contents and their implications for conservation. The new title, "Modeling habitat distribution and niche overlap of Asian horseshoe crabs: Implications for Conservation," explicitly links our research on habitat distribution modeling and niche overlap analysis to their conservation implications (Line 3-5). This revision aims to emphasize our study's contribution to understanding the ecological requirements of Asian horseshoe crabs and providing valuable insights for their conservation and management. We appreciate your attention to detail and your efforts in helping us improve the manuscript.

These claims are not supported by the manuscript, particularly the discussion lacks any contextualisation of results and conservation measures. REformulate the take home message in terms of contribution to gaps in state of the art

Response Thank you for your valuable feedback. We have enhanced both the discussion and conclusion sections with the following content:

Discussion:

To effectively address the pressing conservation challenges outlined above and mitigate the threats to Asian horseshoe crabs, it is imperative to implement a combination of protective measures and sustainable management strategies. The implementation of marine protected areas and monitoring programs is recommended to assess population responses to environmental impacts. Furthermore, regulating the fishing and biomedical harvesting of horseshoe crabs should be strengthened to ensure a balance between conservation and the sustainable use of this biological resource.

Conclusion:

To enhance protection, we recommend establishing marine protected areas (MPAs) in identified critical habitats, particularly focusing on regions where habitat overlap is high, to safeguard shared ecological resources. Additionally, strengthening regulations on the harvest of these species for biomedical purposes is essential to balance conservation with sustainable use. Proactive strategies are needed to assess the impacts of global warming and ocean acidification on horseshoe crab distributions. This includes monitoring potential habitat shifts to new areas due to changing environmental conditions, such as alterations in chlorophyll-a concentrations and wind speed, to inform adaptive conservation measures.

these are repeated from the title, make them more useful to target readers by choosing other words, perhaps focusing on different info

Response Thank you for your valuable feedback. Following the suggestion of another reviewer, we have updated the title to: "Modeling habitat distribution and niche overlap of Asian horseshoe crabs: Implications for Conservation". Additionally, to avoid repetition between the title and the keywords, we have replaced the keywords "Tachypleus tridentatus, Tachypleus gigas, Carcinoscorpius rotundicauda, Potential habitats, Niche overlap, Biodiversity" with "Horseshoe crabs, Species distribution models, Key environmental factors, Niche, Biodiversity" (Line 57-58). These revisions aim to enhance clarity and focus on the core aspects of our study. We appreciate your time and consideration.

The introduction does not clearly position the study contribution in relation to the state of the art for the tested aspects. a great focus should be given to that here.

Response Thank you for your valuable feedback. In response to your comments, we have revisited the introduction section to more clearly articulate the contribution of our study in relation to the current state of the art. Specifically, in Lines 122-136, we have provided a detailed discussion on the advantages of MaxEnt technology, its application in horseshoe crab research, and the existing limitations. Furthermore, we would like to highlight the main contribution of our research. We have overcome the limitations of using conventional low-relevance climate data by incorporating marine environmental parameters, including physical, chemical, and nutritional variables, for the first time in horseshoe crab distribution prediction. This approach provides essential data for their conservation. We believe these revisions have strengthened the manuscript and made our contribution more evident. Thank you again for your helpful suggestions.

These expected contributions do not match the discussion and conclusion contents. delete of change to match.

Response Thank you for your valuable feedback. We have carefully revised the text to ensure that the expected contributions align with the discussion and conclusion contents. We have deleted the sections that were not strongly relevant and made necessary modifications to better highlight the contributions and focus of our study.

The revised text now reads as follows:

As horseshoe crabs are typical intertidal benthic organisms, we hypothesize that water depth, land distance, and specific environmental factors are the primary determinants of the distribution of the three Asian horseshoe crab species. We use the MaxEnt model to predict potential habitats and analyze niche overlap, aiming to provide insights for horseshoe crab conservation. This study also offers methods and ideas for the conservation research of other endangered species, contributing to the advancement of global biodiversity conservation efforts (Line 133-145).

We believe these revisions have strengthened the manuscript and made our contributions more evident. Thank you again for your helpful suggestions.

it is very important to clarify here if the crabs are adults, or different life stages as that should influence results interpretation.

Response Thank you for your insightful feedback. We have taken care to specify in the manuscript that the horseshoe crabs examined in our study are adults (Line 149). This clarification is crucial, as it directly impacts the interpretation of our findings and ensures their accuracy. To prevent any ambiguity, we have explicitly highlighted this detail in the pertinent sections of the text, thereby providing a transparent understanding of the life stage of the horseshoe crabs under investigation. Your constructive suggestions have been invaluable in refining our work.

This subsection needs to explain if food availability was considered, in my interpretation it was not, and if not, why as it is a key distribution factor for any type of crab. Then, introduce the same food availability factor in the discussion (currently it is very incipient therein)

Response Thank you for your valuable feedback. In response to your comments, although food availability was not directly considered, we have included environmental factors such as chlorophyll, nutrients, primary productivity, and organic carbon in our study (Table 1), which serve as indirect indicators of food supply. These factors are closely related to the food resources available to horseshoe crabs and thus can reflect the influence of food availability on their distribution. Furthermore, we have added a detailed explanation regarding food availability in the discussion section, particularly in Lines 472-475, to address this important aspect of their distribution. We believe these revisions have strengthened the manuscript and provided a more comprehensive understanding of the factors influencing horseshoe crab distribution. Thank you again for your helpful suggestions.

This text should be deleted as: 1. its content is more suitable in the introduction section and, 2. it' already given in previous sections. the Discussion should start with interpretation and contextualisation of main results.

Response Thank you for your valuable feedback. We have deleted the text as requested.

this mechanism needs further clarification on how food can vary with depth and known crab responses to food availability

Response Thank you for your suggestions. We have added further explanations on how food varies with depth and known horseshoe crabs responses to food availability as requested. The specific additions are as follows:

Water depth affects the distribution and abundance of horseshoe crab food sources such as plankton and benthic organisms. For example, in shallow water areas with sufficient light, the growth of phytoplankton is promoted, which in turn provides food sources for benthic organisms and indirectly affects the food supply of horseshoe crabs. Additionally, horseshoe crabs migrate annually in response to seasonal and water temperature changes. Every November, they swim from shallow seas to deep-water areas for overwintering; the following April to May, they return to shallow seas for reproductive migration. During migration, horseshoe crabs move between regions with different water depths and obtain different food sources. For example, while overwintering in deep-water areas, they may feed on certain deep-sea organisms; after returning to shallow seas, their food sources change again.

The paragraphs are too long and very difficult to follow from the readers' point of view; create breaks when the topic changes such as here. apply this logic everywhere in the doc.

Response Thank you for your valuable feedback. In response to your concern about the lengthy paragraphs being difficult to follow, I have carefully revised the text by dividing the original paragraph into four thematic sections to enhance clarity and readability (Line 437-522).

This is a key discussion point that requires much more explanation for the entire study. in fact, the methods do not fully clarify if the data obtained in databases corresponds to adults or larvae and what are the implications of it. as stated here, larvae are likely to have different environmental requirements from adults. in other words, are the results being interpreted as simultaneously appl

---

## [Decision Letter · Decision Letter 1]

25 Apr 2025

Modeling habitat distribution and niche overlap of Asian horseshoe crabs: implications for conservation

PONE-D-25-08646R1

Dear Dr. Wang,

We’re pleased to inform you that your manuscript has been judged scientifically suitable for publication and will be formally accepted for publication once it meets all outstanding technical requirements.

Kind regards,

Phuping Sucharitakul

Academic Editor

PLOS ONE

Additional Editor Comments (optional):

Reviewers' comments:

Reviewer's Responses to Questions

**Comments to the Author**

1. If the authors have adequately addressed your comments raised in a previous round of review and you feel that this manuscript is now acceptable for publication, you may indicate that here to bypass the “Comments to the Author” section, enter your conflict of interest statement in the “Confidential to Editor” section, and submit your "Accept" recommendation.

Reviewer #1: All comments have been addressed

Reviewer #2: All comments have been addressed

Reviewer #3: All comments have been addressed

2. Is the manuscript technically sound, and do the data support the conclusions?

Reviewer #1: (No Response)

Reviewer #2: Yes

Reviewer #3: Yes

3. Has the statistical analysis been performed appropriately and rigorously? 

Reviewer #1: (No Response)

Reviewer #2: Yes

Reviewer #3: Yes

4. Have the authors made all data underlying the findings in their manuscript fully available?

Reviewer #1: (No Response)

Reviewer #2: Yes

Reviewer #3: Yes

5. Is the manuscript presented in an intelligible fashion and written in standard English?

Reviewer #1: (No Response)

Reviewer #2: Yes

Reviewer #3: Yes

6. Review Comments to the Author

Reviewer #1: all comments addressed suitably

Reviewer #2: There are a few grammatical problems in the text, but they are too minor to mention here. The authors did a good job, and this article will be very useful and appreciated by readers in many areas of knowledge.

Reviewer #3: The authors have done a good job revising their paper to address my and other reviewers' comments. I believe it can now be accepted for publication as is, and may become an important and highly cited work.

7. PLOS authors have the option to publish the peer review history of their article (what does this mean? ). If published, this will include your full peer review and any attached files.

**Do you want your identity to be public for this peer review?** For information about this choice, including consent withdrawal, please see our Privacy Policy .

Reviewer #1: **Yes: ** Ana Silva

Reviewer #2: **Yes: ** Marcelo Antonio Amaro Pinheiro

Reviewer #3: **Yes: ** Brady K. Quinn

---

## [Editor Report · Acceptance letter]

PONE-D-25-08646R1

PLOS ONE

Dear Dr. Wang,

I'm pleased to inform you that your manuscript has been deemed suitable for publication in PLOS ONE. Congratulations! Your manuscript is now being handed over to our production team.

Kind regards,

on behalf of

Dr. Phuping Sucharitakul

Academic Editor

PLOS ONE